evolution, behaviour, genetics

dispersal, social polymorphism, queen number, supergene, sex ratio, *Formica selysi*

**Authors for correspondence:**
Amaranta Fontcuberta
e-mail: amaranta.fontcuberta@unil.ch
Michel Chapuisat
e-mail: Michel.Chapuisat@unil.ch

†Co-first authors.

# Disentangling the mechanisms linking dispersal and sociality in supergene-mediated ant social forms

Amaranta Fontcuberta†, Ornela De Gasperin†, Amaury Avril, Sagane Dind and Michel Chapuisat

Department of Ecology and Evolution, University of Lausanne, 1015 Lausanne, Switzerland

AF, 0000-0003-1624-4192; MC, 0000-0001-7207-199X

The coevolution between dispersal and sociality can lead to linked polymorphisms in both traits, which may favour the emergence of supergenes. Supergenes have recently been found to control social organization in several ant lineages. Whether and how these 'social supergenes' also control traits related to dispersal is yet unknown. Our goal here was to get a comprehensive view of the dispersal mechanisms associated with supergene-controlled alternative social forms in the ant *Formica selysi*. We measured the production and emission of young females and males by single-queen (monogyne) and multiple-queen (polygyne) colonies, the composition of mating aggregations, and the frequency of crosses within and between social forms in the wild. We found that males and females from alternative social forms did not display strong differences in their propensity to leave the nest and disperse, nor in their mating behaviour. Instead, the social forms differed substantially in sex allocation. Monogyne colonies produced 90% of the females flying to swarms, whereas 57% of the males in swarms originated from polygyne colonies. Most crosses were assortative with respect to social form. However, 20% of the monogyne females did mate with polygyne males, which is surprising as this cross has never been found in mature monogyne colonies. We suggest that the polygyny-determining haplotype free rides on monogyne females, who establish independent colonies that later become polygyne. By identifying the steps in dispersal where the social forms differ, this study sheds light on the behavioural and colony-level traits linking dispersal and sociality through supergenes.

## 1. Background

Sociality and dispersal coevolve. Theory generally predicts that more philopatric phenotypes are selected to cooperate more than dispersing ones [1–4]. This correlation has been found in a large variety of organisms, including bacteria [5], algae [6], fishes [7] and mammals [8,9]. A recent model suggests that the coevolution of social organization and dispersal can lead to linked polymorphisms in both traits, with a cooperative 'non-dispersive' morph and a 'self-serving' dispersive morph [10]. This model predicts that social and dispersal-related traits will be genetically linked, possibly leading to the emergence of supergenes (i.e. clusters of linked loci; [11–13]). Accordingly, recent research has revealed that social organization is controlled by supergenes in at least three independent ant lineages [14]. Each lineage contains socially polymorphic species, and the genotype at a supergene determines whether colonies are headed by one or by multiple queens [15,16]. But whether, and how, 'social supergenes' control dispersal, as predicted by theory, is yet unknown.

Single-queen (monogyne) and multiple-queen (polygyne) ant colonies are often associated with differences in dispersal strategies across, and sometimes within, species [17]. Ant colonies are generally sessile, being composed of mated queens and workers occupying a fixed nest. The colonies produce

winged females and males—the future reproductive individuals. Females mate with one or with multiple males at the beginning of their lives and store the sperm that they will use throughout their lives. Typically, the polygyne social form is more philopatric, producing queens that disperse by foot accompanied by sister-workers, or that stay in their natal nest, whereas the less philopatric, monogyne social form produces queens that disperse on the wing and start colonies independently [18,19]. In several socially polymorphic ant species, stronger population genetic structure in the polygyne social form suggests more restricted dispersal of females, compared to the monogyne social form [20–23]. However, the steps in dispersal at which genetically determined social forms differ remain little studied.

The dispersal of genetically determined social forms is a complex process that involves individual- and colony-level traits. Sociality and dispersal may be correlated at one or several steps. First, colonies may differ in the number of females and males produced. Second, these females and males may vary in their propensity to fly out of the colony and to engage in mating swarms. Third, they may also differ in mating success and ability to establish a new colony. Furthermore, in species with genetic determination of social organization, the phenotype (i.e. monogyne or polygyne) of the resulting colony may depend on allelic dominance and on the frequency of intra- and inter-social form crosses. Yet, most studies have investigated one or few dispersal-related traits, such as colony sex ratio [24,25], individual's propensity to disperse [23], flight behaviour [26–28] or colony founding success [29,30]. To understand how alternative social forms propagate, and how supergenes controlling sociality affect dispersal, one should consider the entire dispersal process.

Here, we investigated (i) whether sociality and dispersal are correlated, as predicted by theory, and (ii) at which specific step(s) does dispersal differ between alternative social forms. We used the socially polymorphic Alpine silver ant, *Formica selysi*, to answer these questions. Most well-sampled populations of this species have both monogyne (single-queen) and polygyne (multiple-queen) colonies [31,32]. Colony social organization is controlled by a large supergene with two non-recombining haplotypes, $M$ and $P$ (previously called Sm and Sp, respectively, [16,33]). Queen and workers in monogyne colonies are homozygous for the $M$ haplotype, whereas queens and workers in polygyne colonies are homozygous for the $P$ haplotype or heterozygous [16]. The $P$ haplotype is dominant, so that heterozygous individuals live exclusively in multiple-queen colonies. In addition, the $P$ haplotype is a transmission ratio distorter causing maternal effect killing [34]. Therefore, polygyne colonies produce exclusively $P$ males, $M/P$ females and $P/P$ females, but no $M$ males nor $M/M$ females.

Previous data suggest that the two social forms differ in some dispersal-related traits. For example, in any given year a higher fraction of monogyne colonies than of polygyne colonies produce winged sexuals, even though both social forms produce winged females and males [35,36]. Monogyne colonies generally produce either females or males (split sex ratio), whereas the sex ratio is more balanced in polygyne colonies [36]. Although social forms show no strong differences in genetic structure or isolation by distance among colonies within populations [31,33], three lines of evidence suggest that dispersal is more restricted in the polygyne form. First, relatedness patterns indicate that polygyne queens are more philopatric and mate with more related males than monogyne

queens [33]. Second, the between-population genetic structure is stronger for the polygyne social form [37]. Third, polygyne queens are less successful at founding colonies independently in laboratory conditions [30,38,39].

Other potentially relevant dispersal-related traits remain puzzling. There is complete assortative mating with respect to social form in mature colonies of the monogyne social form, and partial assortative mating in the polygyne social form [16,33]. Mature monogyne colonies headed by one $M/M$ queen mated to a $P$ male have never been detected in the field [16,33]. Yet, monogyne females did not discriminate against $P$ males in mate choice experiments [39], this cross produces viable offspring in the laboratory [30,38,39], and there is little genetic differentiation between social forms outside of the supergene, which is consistent with ongoing gene flow between social forms [16,31,33,40]. It remains unknown whether the cross between $M/M$ females and $P$ males actually occurs and yields successful colonies in the field. It is possible that $P$ males are rare, do not fly out of their nest or do not meet $M/M$ females in mating swarms, due to temporal or spatial segregation.

To gain insights into how supergene-controlled social forms disperse in the wild, we investigated actual flight and mating by each sex from both social forms of the Alpine silver ant, *F. selysi*. We assessed the production and emission of winged females and males by monogyne and polygyne colonies, the composition of mating swarms in terms of sex and social origin, and the frequency of intra-form and inter-form matings in swarms. In specific, we investigated (i) whether monogyne and polygyne colonies differ in their contribution of females and males to mating swarms, and if it is due to differences in colony sex allocation and/or in the propensity of sexuals to disperse, and (ii) whether the actual mating pattern in swarms is assortative and matches the observed pattern in mature field colonies. We specially examined if females of monogyne origin mate with males of polygyne origin, or whether there is spatial or temporal segregation between social forms. By identifying at which steps in the dispersal process do supergene-controlled ant social forms differ, this study sheds light on the mechanisms genetically linking dispersal and sociality.

## 2. Methods

### (a) Mating swarms sampling

We sampled *F. selysi* females and males in field mating swarms. On sunny days, winged females and males take off from their natal colonies and fly to the top of birch, pine and spruce trees, where they mate on branches. We looked for swarms in two Swiss populations, Derborence (7°12′56″ E, 46°16′50″ N, altitude 1450 m) and Finges (7°36′30″ E, 298 4°18′30″ N, altitude 565 m), over three consecutive reproductive seasons (June–August 2016–2018). The two populations are approximately 50 km away from one another. Each population consists of hundreds of colonies distributed in large patches of mosaic, floodplain habitat [32,41]. Females and males captured in swarms emerged from colonies of each local population, of which a sample was monitored for sexual production (see below). We observed mating swarms between 09.00 and 13.00 on 20 days (electronic supplementary material, table S1). We captured females and males flying around tree branches, using butterfly nets mounted on 4-m long poles. We also collected females standing or mating on tree branches, walking alone or being pulled by workers on

the ground. The females on the ground had lost their wings and were searching for a nest site, after having participated in swarms (electronic supplementary material, tables S1 and S2).

Males were stored individually in EtOH 70%. Females were kept in rearing units composed of a glass tube wrapped in aluminium foil, with water retained by a cotton plug at the bottom, which mimics independent claustral colony founding by queens after mating [30]. We recorded the number of workers produced by each queen after approximately six weeks in the laboratory, when the first cohorts of eggs had developed into adult workers. At this point, we determined the mating status of the queens and collected the sperm of their mates for later genotyping. We dissected the abdomen of each queen, extracted the sperm of their spermathecas, if any, and stored the sperm in Qiagen ATL buffer with proteinase K.

## (b) Emission and production of females and males
We monitored the emission of females and males by 32 monogyne colonies (18 in Derborence and 14 in Finges) and 14 polygyne colonies (12 in Derborence and two in Finges) over the entire 2016 reproductive season. We identified colonies that were producing winged females and males (= alates) by noting the presence of sexual larvae or pupae under flat stones covering nest entrances. We placed emergence tents (MegaView Science Co., Taiwan) over these colonies when the first alates were observed underneath stones. We sampled all dispersing alates by visiting the tents every 3 days, until no more alates were collected for two consecutive days in any colony (i.e. from 21 June to 4 August in Finges and from 26 July to 21 September in Derborence). To verify that the females emitted by colonies were virgin, we dissected the spermatheca of 36 females from 13 monogyne colonies and 25 females from eight polygyne colonies.

We monitored the production of sexual pupae and alates in 22 monogyne colonies and 19 polygyne colonies in Derborence. We turned the nest-covering stones and sampled sexual pupae two to three times in each colony, at least one week apart, from 26 June to 3 August 2018. We collected as many pupae as possible, alongside 20 workers, each time. We reared pupae with nest-mate workers in the laboratory and counted the number of males and females that hatched.

## (c) Supergene genotyping
The social structure of colonies, the social origin of females and males sampled in or after swarms and the social origin of the sperm stored in the females' spermatheca were determined by genotyping SNPs diagnostic for alternative haplotypes at the social supergene [16,33]. We extracted DNA from the thorax of females and males, using Qiagen DNeasy mini-spin columns (for samples collected in 2016) or Qiagen BioSprint 96 Workstation (for samples collected in 2017 and 2018). DNA was extracted from sperm using Qiagen DNeasy mini-spin columns. DNA from three workers per field colony was extracted from a single leg placed in a 6% Chelex solution with proteinase K. For samples collected in 2016 and 2017, the genotyping of haplotype-specific SNPs was done with a PCR-RFLP assay, as described previously [16]. For samples collected in 2018, we used a novel qPCR assay. The qPCR assay was developed by aligning RAD-sequencing data from workers (Avril et al., [33]). For designing the TaqMan probes, we selected a conserved region of the supergene possessing three haplotype-specific SNPs. The qPCR contained 1X TaqPath ProAmp Master Mix (Thermo Fisher Scientific), 100 nM of each haplotype-specific TaqMan probe (Microsynth CH) and 200 nM of each primer (Microsynth, CH). We used the following primers and probes: forward primer: TCGCGCAATTATCTCGTCTA; reverse primer: TGATAGCGGCATCAATCTACA; 'M'-specific TaqMan probe: FAM_TTCACTCCTCCACAAGAGAA_MGB-Q500; 'P'-specific

TaqMan probe: ATTO550_TTTGCTTCTCCACAAGAGAA_MGB-Q500. The reaction was carried in a final volume of 20 µl, from which 4 µl was the extracted DNA template. The qPCR cycling conditions were: 60°C for 1 min; 95°C for 5 min; 40 cycles of 95°C for 15 s and 60°C for 1 min; and 60°C for 1 min.

Polygyne colonies do not produce $M/M$ females or $M$ males [16,33,34]. To confirm complete transmission ratio distortion towards $P$, we genotyped 180 males captured in emergence tents placed over 12 polygyne colonies (15 males per colony). As expected, all these males carried exclusively the $P$ haplotype. Conversely, the $P$ genotype is absent from monogyne colonies [16,33]. Together, these data show that $M/M$ females and $M$ males captured in swarms originated from monogyne colonies, while $P/M$ or $P/P$ females and $P$ males originated from polygyne colonies.

## (d) Statistical analyses
### (i) Assortative mating
We tested whether females captured in swarms mated more frequently with males of their own or of the alternative social origin (determined by the sperm genotype). We ran a generalized linear model (GLM) with binomial error distribution, where the response variable was the female's mate type (same or alternative social form). We included as explanatory variable 'population' (Derborence or Finges). If the probability to mate with males of the same social form is higher than random, we expect the intercept to be significantly larger than 0.5. Next, we compared whether the proportion of monogyne females mated to each type of male was different from the proportion of males of each social form in swarms (we did not test for assortative mating in polygyne females due to low sample size). We ran a GLM with binomial error distribution, where the response variable was the social form of each male. We included as explanatory variables 'stage' (sperm or swarms) and 'population' (Derborence or Finges). The interaction between the two factors was not significant (estimate = −1.05, s.e. = 0.7, $z = −1.5$, $p = 0.14$) and was removed when estimating the main terms. There was neither over-dispersion nor non-uniformity of residuals in either model.

### (ii) Emission of females and males
To test if the number of females and males captured in emergence tents differed between social forms, we ran separate models with total number of females and total number of males emitted per colony as response variable, respectively. The small sample size in Finges precluded us from comparing populations. We analysed all colonies together and re-run the analyses with colonies from Derborence only, which confirmed that population differences were not biasing the results. We ran zero-inflated negative binomial (ZINB) regression models [42]. We included as explanatory variable the social form of each colony in the count component of the models (i.e. number of males or females emitted) and we included only the intercept in the zero component of the models (i.e. likelihood of emitting zero males or females). There was neither over-dispersion nor non-uniformity in the residuals in either of the two models.

Colony sex ratio was calculated as the proportion of females among alates emitted or produced by each colony. Colonies emitting more than 90% of females were classified as 'female-specialists', while colonies producing more than 90% of males were classified as 'male-specialists' [36]. To examine whether females from alternative social forms differ in their propensity to disperse, we compared the colony sex ratios of alates produced and emitted in the population Derborence. If a large fraction of the females produced by polygyne colonies do not disperse, the sex ratio of alates emitted should be more male-biased than the sex ratio of alates produced. To compare colony sex ratio

**Table 1.** Social composition and actual crosses in swarms. The social origin of males and females participating in mating swarms was inferred from their social supergene genotypes. The social origin of the females' mates was inferred by genotyping the sperm in their spermatheca.

| | monogyne origin | polygyne origin | |
|---|---|---|---|
| **males** | | | |
| *supergene haplotype* | M | P | |
| number of males | 345 (43%) | 458 (57%) | |
| **females** | | | |
| *supergene genotype* | M/M | M/P | P/P |
| mated to M male | 52 | 0 | 1 |
| mated to P male | 13 | 2 | 0 |
| mated to undetermined male | 12 | 3 | 1 |
| virgin | 9 | 1 | 2 |
| total number of females | 86 (89.6%) | 6 (6.2%) | 4 (4.2%) |

between the two stages ('produced', i.e. sampled under stones, versus 'emitted', i.e. captured in emergence tents) and between social forms, we ran a generalized linear mixed model (GLMM) with binomial error distribution. The response variable was the weighted proportion of females in each colony. We included 'stage' and 'social form' as explanatory variables, and the colony of origin as a random effect. We also included an observation level random effect (OLRE) to account for over-dispersion [43]. The estimates for the main terms were obtained after removing the interaction from the model. There was neither over-dispersion nor non-uniformity of residuals in the resulting model.

### (iii) Observed and expected composition of swarms

We calculated the expected composition of mating swarms in Derborence by multiplying the average number of females and males emitted by colonies of one social form by the proportion of colonies of this social form among colonies producing alates in this population. The expected proportion of each social form in swarms was calculated separately for each sex, using the following formula:

(1) expected proportion of females (or males) of monogyne origin

$$= \frac{M * Nm}{M * Nm \; + \; (1 - M) * Np},$$

where $M$ is the proportion of monogyne colonies among colonies producing alates in Derborence (0.61, averaged over three field seasons 2016, 2017 and 2018) and $Nm$ and $Np$ are the average number of females (or males) emitted by monogyne and polygyne colonies, respectively.

We then examined whether the observed composition of mating swarms (i.e. the proportions of females and males of monogyne and polygyne origin in swarms in Derborence) matched this expected composition. All analyses were performed in R v. 3.4.4 [44]. Generalized linear models were fitted using the 'glmmTMB' function [45]. Over-dispersion and non-uniformity of residuals were evaluated with the 'DHARMa' R package [46].

## 3. Results

### (a) Mating swarms

We captured 96 females and 803 males in mating swarms (electronic supplementary material, table S1). The vast majority of the females belonged to the monogyne social form (89.6%; table 1; total $N = 96$). By contrast, many males belonged to

the polygyne social form (57%; table 1; $N = 803$). Females of monogyne and polygyne origin were sampled while performing similar behaviours (electronic supplementary material, table S2). The majority of the females captured were inseminated (table 1).

### (b) Actual crosses in swarms

The genotyping of the sperm stored in the spermatheca of mated females revealed that all four possible types of crosses naturally occurred in mating swarms. Specifically, females belonging to each social form had mated with males from the same social form and with males from the alternate social form (table 1). Most matings were assortative with respect to social form (proportion of assortative mating = 79.4%, $N = 68$; estimate *intercept* $= 1.6$, s.e. $= 0.41$, $z = 3.89$, $p < 0.001$). There was no significant difference between populations (estimate 'Finges' $= -0.61$, s.e. $= 0.61$, $z = -1.01$, $p = 0.31$; table 1). Overall, 80% of all mated females of monogyne origin had mated with males of monogyne origin ($N = 65$; table 1). The proportion of assortative mating by monogyne females was higher than expected given the proportion of monogyne males in swarms (43%, $N = 803$; table 1; estimate 'swarms' $= 1.54$, s.e. $= 0.33$, $z = 4.68$, $p < 0.001$), independently of the population (estimate 'Finges' $= -0.44$, s.e. $= 0.28$, $z = -1.56$, $p = 0.12$). Two out of three females of polygyne origin had mated with males of polygyne origin (table 1).

All four types of crosses yielded viable offspring. After six weeks, 46 out of 50 females of monogyne origin had workers, as well as two out of three females of polygyne origin. The number of workers produced by monogyne females was not significantly related to the social origin of their mates ($N = 45$; Wilcoxon rank-sum test, $W = 149.5$, $p = 0.98$; electronic supplementary material, figure S1).

### (c) Emission of females and males

Over the entire reproductive season, we captured 4505 winged females and 13 360 males in emergence tents placed on top of field colonies. The vast majority of the emerging females were virgin. Out of 61 females, only one, from a polygyne colony, was inseminated.

Colony productivity and sex ratio differed markedly between social forms. On average, a monogyne colony emitted

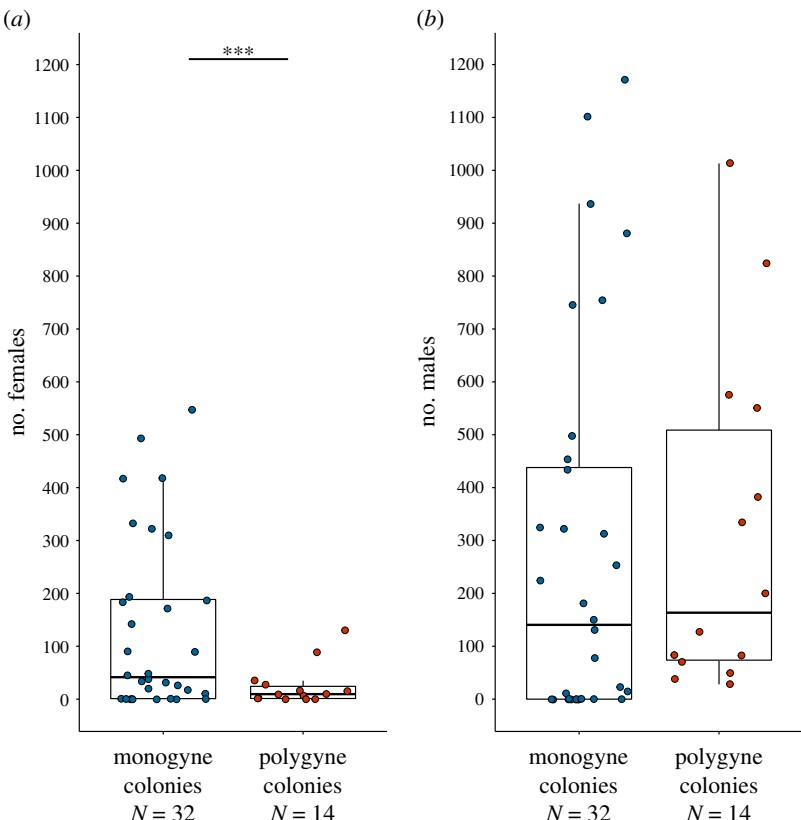

**Figure 1.** Emission of females (*a*) and males (*b*) by monogyne (blue dots) and polygyne (red dots) colonies. The bold horizontal line is the median, the lower and upper hinges correspond to the first and third quartiles, and whiskers extend to 1.5 times the inter-quartile range. ***stands for *p*-value < 0.001. (Online version in colour.)

5.3 times more females than a polygyne colony (mean per monogyne colony = 130.2, s.d. = 164.5; mean per polygyne colony = 24.4, s.d. = 38.4; estimate 'polygyne' = −1.7, s.e. = 0.47, $z = -3.7$, $p < 0.001$; figure 1). This difference was still highly significant when considering only the main population Derborence (estimate 'polygyne' = −2.22, s.e. = 0.49, $z = -4.52$, $p < 0.001$). Monogyne and polygyne colonies emitted on average similar numbers of males (mean per monogyne colony = 281.2; mean per polygyne colony = 311.6; estimate 'polygyne' = −0.31, s.e. = 0.35, $z = -0.9$, $p = 0.37$; when considering Derborence only: estimate 'polygyne' = 0.019, s.e. = 0.43, $z = 0.043$, $p = 0.97$; figure 1).

The proportion of female-specialist and male-specialist colonies differed between social forms (Fisher's exact test, $p = 0.006$). Colony sex ratio was split among monogyne colonies, with 40.6% of the colonies specializing in the emission of females and 43.8% specializing in the emission of males. By contrast, all polygyne colonies emitted a majority of males, with 78.6% being male-specialists and the rest having moderately male-biased sex ratios (figure 2). Colony sex ratio in Derborence differed between social forms (GLMM, estimate 'polygyne' = −4.41, s.e. = 1.22, $z = -3.63$, $p < 0.001$). Likewise, colony sex investment differed markedly between social forms (see electronic supplementary material, figure S2).

The sex ratio in alates emitted was not significantly different from the sex ratio in alates produced (estimate 'produced' = 0.63, s.e. = 0.89, $z = 0.7$, $p = 0.48$), independently of the social form (estimate interaction 'stage' × 'social form' = −1.2, s.e. = 1.9, $z = -0.66$, $p = 0.51$). All polygyne colonies produced and emitted a majority of males, whereas approximately half of the monogyne colonies produced and emitted a majority of females (figure 2; electronic

supplementary material, figure S3). This indicates that the low number of females emitted by polygyne colonies is mainly due to the low number of females produced, rather than to females not leaving the colony on the wing.

At the population level, females and males from both social forms were emitted in synchrony (electronic supplementary material, figure S4). Each colony emitted females or males on several peak days, over a period of about 1.5 months. Overall, the main flight peaks occurred at the same time for monogyne and polygyne colonies (electronic supplementary material, figure S4).

### (d) Observed versus expected composition of swarms

The proportion of males of polygyne origin observed in mating swarms in Derborence was slightly higher than expected given the emission of males by polygyne and monogyne colonies in this population (swarms: 57.9% of the males were of polygyne origin; expected proportion: 46.9%; goodness-of-fit $\chi^2$ test, $X^2 = 36.65$, d.f. = 1, $p < 0.001$). The same trend was observed for females (swarms: 13.7% of the females were of polygyne origin; expected proportion: 6.5%; binomial exact test, $p = 0.038$).

## 4. Discussion

Theory suggests that alternative dispersal and social strategies coevolve and may become linked within alternative supergene haplotypes [1,2,10]. Yet experimental proof that dispersal-related traits are correlated to alternative forms of social organization determined by supergenes is scarce. In the few documented cases where variation in social

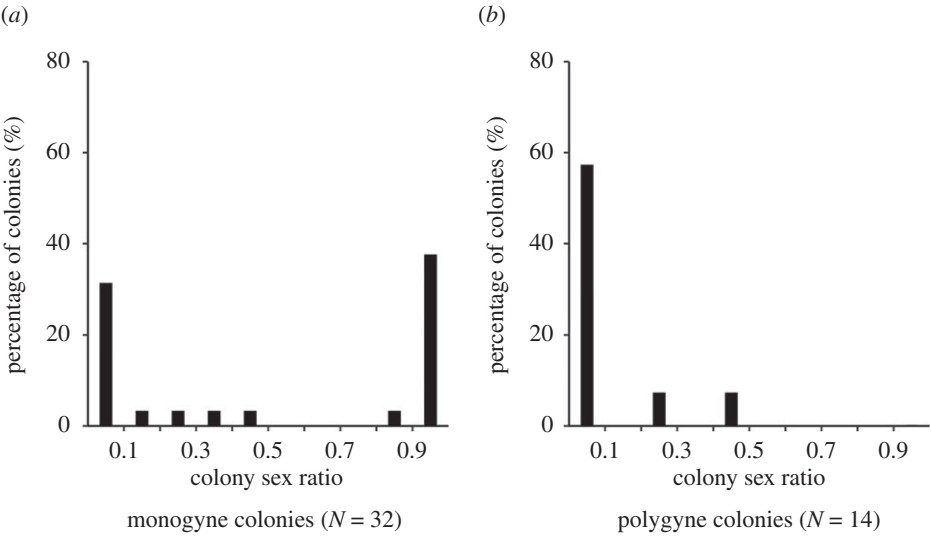

**Figure 2.** Colony sex ratio (proportion of females emitted) of monogyne (*a*) and polygyne (*b*) colonies. Bars represent the percentage of colonies belonging to the corresponding colony sex ratio class. The figure includes colonies from the Derborence and Finges populations.

behaviour and dispersal are genetically associated, individuals that are more aggressive typically show higher propensity to disperse (e.g. [47,48]). But in eusocial species, where this genetic link is likely to evolve, dispersal may not only be affected by individual traits, but also by colony-level phenotypic traits. Moreover, the spread of genetically determined social forms may also depend on properties of the underlying genetic system.

Here, we show that supergene-controlled alternative social forms differ in dispersal, as predicted by theory, but through unexpected mechanisms. The differences in dispersal between genetically determined social forms of the Alpine silver ant, *F. selysi*, are largely due to colony-level traits such as the production of dispersers (winged females and males), rather than to individual-level traits, such as flight and mating behaviour. The monogyne social form produces the large majority of disperser females. By contrast, the more philopatric polygyne social form specializes in the production of disperser males. The asymmetrical pattern of sex allocation, coupled to dominance and transmission ratio distortion by the polygyny-associated haplotype, suggests that the polygyne social form might mostly spread through males of polygyne origin mated to females of monogyne origin.

Females and males from each social form engaged in dispersal flights and joined mating swarms. Yet, the social forms differed strikingly in their contribution to disperser females. As many as 90% of the females sampled in mating swarms came from monogyne colonies, while 10% came from polygyne colonies. To find out whether the rarity of polygyne females in swarms was caused by low female production or by a propensity of females to stay in the nest, we compared the colony sex ratios of alates produced and emitted. We did not detect any significant difference between the two stages, which suggests that the rarity of polygyne females mainly originates at the production stage. Thus, the strong difference between the monogyne and polygyne social form in their contribution to dispersing females is due to pronounced differences in colony sex allocation, rather than to the flight propensity of females.

The sharp contrast in sex allocation between social forms and the split sex ratio within the monogyne social form raise questions about the underlying mechanisms. Recently, colony

sex allocation in *Formica glacialis* has been shown to be affected by a supergene adjacent to the one controlling social organization [49]. The differences in sex allocation between *F. selysi* social forms could also be under direct or indirect control from the social supergene. Very diverse supergene-mediated effects are possible, from social discrimination [50] to transmission ratio distortion [34,51] or caste distortion [52]. Regarding the split sex ratio within the monogyne social form, previous studies have shown that queens largely control colony sex ratio in *F. selysi* [36] and *Solenopsis invicta* [53], by biasing the sex ratio in their eggs. The proximate factors by which monogyne queens specialize in laying fertilized or unfertilized eggs remain to be investigated.

Females and males from the polygyne form displayed similar flight and mating behaviour than their counterparts from the monogyne form. Males from both social forms were much more abundant than females, both in swarms and when emitted by field colonies. Polygyne males were even more abundant in swarms than expected given the number of males emitted by polygyne colonies. Similarly, and contrary to the common view that polygyny is associated with reduced dispersal by flight and intranidal mating, we found that a high proportion of the rare polygyne females disperse on the wing, participate in swarms, and show similar mating behaviour to monogyne females. After their dispersal flight, polygyne females might attempt independent colony founding. In line with this hypothesis, laboratory experiments showed that *F. selysi* polygyne females succeeded at founding colonies independently, although less successfully than monogyne females [30,38,39]. Thus, dispersal through mated females is one of the possible mechanisms by which the polygyne social form can propagate, although it is unlikely to represent a major dispersal route due to the low number of dispersing females this social form produces.

All four possible crosses, within and between social forms, occurred in nature, but most of the matings were assortative with respect to social form. How this strong assortative mating arises remains unknown, as laboratory experiments have failed to detect female preference for males of their own social form [39]. Interestingly, we found a cross that is absent from extensive surveys of established colonies. Indeed, our sampling and genotyping from swarms demonstrated that

females of monogyne origin did mate with males of polygyne origin in natural conditions. This cross accounted for 20% of the matings by monogyne females in swarms, whereas it has never been detected in mature field colonies [16,33]. This discrepancy raises the question, what is the fate of this cross in nature? Previous studies indicate that genetic incompatibilities are unlikely, since females of monogyne origin experimentally mated to males of polygyne origin produced brood and succeeded at founding incipient colonies in laboratory conditions [30,38,39]. The present study adds further evidence that monogyne females mated with polygyne males in natural field conditions do produce a viable cohort of worker offspring, and they produced as much brood as their counterparts mated to a male of monogyne origin before hibernation. Further investigation is needed to determine whether this cross fails to establish mature colonies in the field. The relative rarity of the cross formed by $M/M$ queens mated to $P$ males, combined with the fact that in the laboratory these colonies are smaller after a year and less competitive than colonies formed by $M/M$ queens mated to $M$ males (O.D.G, 2020, unpublished data), likely contributes to explain why $M/M$ queens mated to $P$ males have not been found in mature field colonies.

Alternatively, or additionally, females of monogyne origin mated to males of polygyne origin may engender polygyne colonies. These females carry the dominant polygyny-associated supergene haplotype, $P$, in their spermatheca [16,33]. Therefore, all workers produced by this cross will carry the $P$ haplotype. These workers are unlikely to adopt lone alien queens [54], but they may adopt additional $P$-carrying sister queens when the colony produces winged females, or queens from nearby polygyne colonies when they are accompanied by workers [54]. The colony may thus become multiple-queened, and as the colony grows and matures, the founding queen may be replaced or remain undetected. This hypothesis would imply that polygyne colonies can propagate by the conversion of incipient monogyne colonies into mature polygyne colonies through male transmission of the $P$ supergene haplotype.

Through the sperm of males hitchhiking in females of monogyne origin, the polygyne social form would profit from the better colonization abilities of these females. This scenario for the spread of the polygyne form has also been proposed for the fire ant $S.$ $invicta$ [55] and may thus be common to socially polymorphic species with a genetic basis of social organization. In mice, individuals carrying the $t$-haplotype were more likely to emigrate and possibly propagate the distorter to populations where it is rare [56]. Similarly, our data in $F.$ $selysi$ suggest that the distorter supergene haplotype $P$ may favour its own propagation to new populations through inter-form crosses. In $F.$ $selysi$, females of monogyne origin are numerous and successful at independent colony founding. Twenty per cent of these females

mated with polygyne males, which are also abundant in swarms. If this cross indeed results in mature polygyne colonies, monogyne females might be the main vehicle of spread for the $P$ supergene haplotype and the associated polygyne social form. As in source–sink dynamics, monogyne females mated to polygyne males may act as source colonizers of distant habitats and serve as bridgehead for the subsequent propagation of the polygyne form.

Altogether, these results illustrate how genetic, behavioural and colony-level factors jointly affect the spread of genetically determined forms of social organization. Monogyne colonies of $F.$ $selysi$ produce a large number of females that disperse on the wing, mate in swarms, and found colonies independently. Polygyne colonies emit few females and many males that also join mating swarms. Mating between social forms readily occurs in the field, and the cross between polygyne males and monogyne females likely favours the spread of the polygyny-associated supergene haplotype, which is dominant and distorts segregation. Thus, the polygyny-associated supergene haplotype may be a free rider exploiting the colony-founding abilities of the monogyne females.

In conclusion, our results provide a clear biological example where dispersal and sociality are genetically associated, a link previously predicted by theory [10]. Moreover, our results show that this link can be achieved through unexpected mechanisms, like colony sex-allocation, and through the outcome of inter-form crosses, and not necessarily through large differences in individual's dispersal and mating behaviours, as it is often implicitly assumed. They, therefore, illustrate how holistic studies that consider different parts of a species' life cycle are required to unravel dispersal differences across genetically determined social forms.

**Ethics.** This work complied with the relevant legal requirements of the University of Lausanne and Switzerland. $F.$ $selysi$ is not an endangered species.

**Data accessibility.** Datasets supporting this article are available from the Dryad Digital Repository: http://doi.org/10.5061/dryad.t76hdr80k [57].

**Authors' contributions.** A.F., O.D.G and M.C. designed the study; A.F., O.D.G and S.D. collected field data; A.F., O.D.G and A.A. performed wet-laboratory analyses; A.F., O.D.G and M.C. analysed the data; A.F., O.D.G and M.C. wrote the manuscript. We declare we have no competing interests.

**Competing interests.** We declare we have no competing interests.

**Funding.** This work was supported by the Swiss National Science Foundation (grant no. 31003A-173189/1).

**Acknowledgements.** We thank Franck Chalard, Santiago Herce Castañón, Lorrain Voisard, Sze Huei Yek and Sacha Zahnd for help in the field; Timothée Fettrelet, Pierre Blacher, Jason Buser and Christophe Lakatos for help in rearing ants and genotyping; Raphael Scherrer for advice on statistics; and Sanja Hakala for comments on the manuscript. We are grateful to Tanja Schwander for fruitful discussions.

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
