## [Peer Review File · Proceedings of the Royal Society B: Biological Sciences]

Review History

RSPB-2021-0118.R0 (Original submission)

Review form: Reviewer 1 (Yannick Wurm)

Recommendation

Accept with minor revision (please list in comments)

Scientific importance: Is the manuscript an original and important contribution to its field?
Good

General interest: Is the paper of sufficient general interest?
Good

Quality of the paper: Is the overall quality of the paper suitable?
Excellent

Is the length of the paper justified?
Yes

Should the paper be seen by a specialist statistical reviewer?
No

Do you have any concerns about statistical analyses in this paper? If so, please specify them explicitly in your report.

No

It is a condition of publication that authors make their supporting data, code and materials available - either as supplementary material or hosted in an external repository. Please rate, if applicable, the supporting data on the following criteria.

Is it accessible?

No

Is it clear?

Yes

Is it adequate?

No

Do you have any ethical concerns with this paper?

No

Comments to the Author

A social supergene system controls whether colonies of *Formica* ants have one or multiple queens. The authors perform extensive field collections and laboratory rearing experiments in order to understand the transmission of the supergene variants. This includes crosses and rearing of queens of the different genotypes with males of the genotypes, population-wide captures of alates leaving their nests, capturing and genotyping of male and queen brood from within nests, and post mating-flight capturing of mated queens and their sperm.

The results read a bit dry - this is not a weakness but rather a necessary strength for fully reporting the different experimental results.

The methods are appropriate, with high sample sizes except when the biology was limiting (e.g., a lack of flying queens from multiple-queen-colonies). Intriguingly, the authors detect that multiple-queen colonies produce almost exclusively males. This differs substantially from single-queen colonies, where colonies tend to be specialised in producing either queens or males. The reasons for this are unknown. What could be the ecological reasons for this? Alternatively, could there be genetic factors affecting sex allocation? There is some evidence of supergene-related sex distortion in the fire ant system (a Ross paper, and Buechel et al *Mol Ecol* 2014).

Crucially (and somewhat counterintuitively), the results suggest that dispersal of polygyne colonies likely occurs through initial establishment of new colonies by monogyne queens, with subsequent conversion of these colonies to polygyny.

The results are of the quality and general relevance that are appropriate for Proc B. The paper stands well as is, thus I feel that I do not need to see a revised version of this paper.

Do the authors have any mitochondrial sequence data that could support the general conclusions of monogyne-queen-mediated polygyne founding?

I have mostly minor suggestions:

- Line 30: citation 15 is unnecessary, as citation 17 cited 2 lines later is the actual paper reporting this. I am also not convinced that an unpublished PhD thesis (citation 16) should be cited.

- Line 73: grammar: "whether" is inappropriate

- supergene genotyping: the authors develop and report a novel genotyping assay, but provide no rationale or mechanism through which it was developed. This likely includes some SNP comparisons? Does it target the same polymorphism as the previous RFLP assay? Could the ongoing low-level recombination observed in another of the authors recent papers be problematic

for this?

- line 229 - given the paucity of polygyne queens ($n = 6$ and $n=4$, compared to 86 monogyne), I am not convinced that this conclusion wants to be drawn.

- The proc b reviewer form indicates "It is a condition of publication that authors make their supporting data, code and materials available" - the data appear to be here, but the code is not provided.

Review form: Reviewer 2

Recommendation

Major revision is needed (please make suggestions in comments)

Scientific importance: Is the manuscript an original and important contribution to its field?

Excellent

General interest: Is the paper of sufficient general interest?

Excellent

Quality of the paper: Is the overall quality of the paper suitable?

Excellent

Is the length of the paper justified?

Yes

Should the paper be seen by a specialist statistical reviewer?

No

Do you have any concerns about statistical analyses in this paper? If so, please specify them explicitly in your report.

No

It is a condition of publication that authors make their supporting data, code and materials available - either as supplementary material or hosted in an external repository. Please rate, if applicable, the supporting data on the following criteria.

Is it accessible?

Yes

Is it clear?

Yes

Is it adequate?

Yes

Do you have any ethical concerns with this paper?

No

Comments to the Author

"Social chromosomes" of ants are an important model system in understanding the evolution of supergenes underlying polymorphisms. This paper focuses on the social supergene in *Formica selysi*, where it is associated with queen numbers of colonies. Population data from mating flights, and colony sex ratio data is used to investigate the association between the supergene genotype and dispersal and mating behaviours. The data shows that the main difference between

the social forms is not in the dispersal and swarming propensity, but in colony sex ratios, and assortative mating. The paper is important, clearly argued and analysed, and the results very interesting. I have some further analyses I would like to see though, as well as some clarifications.

1. As the sex allocation patterns are very interesting, it would be nice to see the colony and population data also presented and analysed as investment ratios. Presumably sexual dimorphism is not very high in this species, so the pattern should remain quite similar. Nevertheless, it would be an interesting complement to the big picture to understand the sex allocation patterns in this respect as well – whose optima is the sex ratio at, at the population level, or within colonies of either morph (and do the split sex ratio patterns make sense in this respect)?

Some further clarifications and explanations I would like to see:

2. A few sentences of explanation about the populations would be helpful: how closed are the populations? Are they clearly lined patches of suitable habitat in a matrix of unsuitable habitat, or more continuous and less clearly defined? If the former, how far are the next populations? Some explanation on this would give the reader an idea of how exclusively the mating flight data corresponds to the colony data.

3. The possible scenario of hitch-hiking of P-males on MM-females, and the turning of M-nests into P via recruiting MP queens is very interesting. But given that such crossings are not found in the years of collections by the authors, some additional explanation would be helpful. This would mean that the colonies recruit new queens, and get rid of the old ones, before they are large enough to be found in field collections, right? But this would mean that they produce sexuals at very small colony sizes, which would be unusual, or that they recruit (or are infiltrated by) queens from the outside, which would be surprising, and would at least temporarily lead to multiple mtDNA lineages within colonies. The other option would be that for some reason the success of such crosses observed in lab rearings is not the case in the wild – if there are any likely scenarios for this, it would be useful to hear. So a few sentences of explanation would be nice.

Decision letter (RSPB-2021-0118.R0)

09-Feb-2021

Dear Dr Fontcuberta:

Your manuscript has now been peer reviewed and the reviews have been assessed by an Associate Editor. The reviewers' comments (not including confidential comments to the Editor) and the comments from the Associate Editor are included at the end of this email for your reference. As you will see, the reviewers and the Editors have raised some concerns with your manuscript and we would like to invite you to revise your manuscript to address them.

When submitting your revision please upload a file under "Response to Referees" - in the "File Upload" section. This should document, point by point, how you have responded to the

reviewers' and Editors' comments, and the adjustments you have made to the manuscript. We require a copy of the manuscript with revisions made since the previous version marked as 'tracked changes' to be included in the 'response to referees' document.

Research ethics:

Use of animals and field studies:

It is a condition of publication that you make available the data and research materials supporting the results in the article. Please see our Data Sharing Policies (<https://royalsociety.org/journals/authors/author-guidelines/#data>). Datasets should be deposited in an appropriate publicly available repository and details of the associated accession number, link or DOI to the datasets must be included in the Data Accessibility section of the article (<https://royalsociety.org/journals/ethics-policies/data-sharing-mining/>). Reference(s) to datasets should also be included in the reference list of the article with DOIs (where available).

Online supplementary material will also carry the title and description provided during submission, so please ensure these are accurate and informative. Note that the Royal Society will not edit or typeset supplementary material and it will be hosted as provided. Please ensure that

the supplementary material includes the paper details (authors, title, journal name, article DOI). Your article DOI will be 10.1098/rspb.[paper ID in form xxxx.xxxx e.g. 10.1098/rspb.2016.0049].

Please submit a copy of your revised paper within three weeks. If we do not hear from you within this time your manuscript will be rejected. If you are unable to meet this deadline please let us know as soon as possible, as we may be able to grant a short extension.

Best wishes,
Professor Gary Carvalho
mailto: proceedingsb@royalsociety.org

Associate Editor

Board Member: 1

Comments to Author:

Your manuscript has received two very positive reviews, with which I concur. As you'll see, there are some requests for clarification and additional text, which you need to address by making appropriate modifications in the manuscript. It's also important that you make the R code available, as highlighted by one of the reviewers. I look forward to the revised version and a cover letter explaining the additions and changes you've made.

Reviewer(s)' Comments to Author:

Referee: 1

Comments to the Author(s)

A social supergene system controls whether colonies of *Formica* ants have one or multiple queens. The authors perform extensive field collections and laboratory rearing experiments in order to understand the transmission of the supergene variants. This includes crosses and rearing of queens of the different genotypes with males of the genotypes, population-wide captures of alates leaving their nests, capturing and genotyping of male and queen brood from within nests, and post mating-flight capturing of mated queens and their sperm.

The results read a bit dry - this is not a weakness but rather a necessary strength for fully reporting the different experimental results.

The methods are appropriate, with high sample sizes except when the biology was limiting (e.g., a lack of flying queens from multiple-queen-colonies). Intriguingly, the authors detect that multiple-queen colonies produce almost exclusively males. This differs substantially from single-queen colonies, where colonies tend to be specialised in producing either queens or males. The reasons for this are unknown. What could be the ecological reasons for this? Alternatively, could there be genetic factors affecting sex allocation? There is some evidence of supergene-related sex distortion in the fire ant system (a Ross paper, and Buechel et al Mol Ecol 2014).

Crucially (and somewhat counterintuitively), the results suggest that dispersal of polygyne colonies likely occurs through initial establishment of new colonies by monogyne queens, with subsequent conversion of these colonies to polygyny.

The results are of the quality and general relevance that are appropriate for Proc B. The paper stands well as is, thus I feel that I do not need to see a revised version of this paper.

Do the authors have any mitochondrial sequence data that could support the general conclusions of monogyne-queen-mediated polygyne founding?

I have mostly minor suggestions:

- Line 30: citation 15 is unnecessary, as citation 17 cited 2 lines later is the actual paper reporting this. I am also not convinced that an unpublished PhD thesis (citation 16) should be cited.

- Line 73: grammar: "whether" is inappropriate
- supergene genotyping: the authors develop and report a novel genotyping assay, but provide no rationale or mechanism through which it was developed. This likely includes some SNP comparisons? Does it target the same polymorphism as the previous RFLP assay? Could the ongoing low-level recombination observed in another of the authors recent papers be problematic for this?
- line 229 - given the paucity of polygyne queens (n = 6 and n=4, compared to 86 monogyne), I am not convinced that this conclusion wants to be drawn.
- The proc b reviewer form indicates "It is a condition of publication that authors make their supporting data, code and materials available" - the data appear to be here, but the code is not provided.

Referee: 2

Comments to the Author(s)

"Social chromosomes" of ants are an important model system in understanding the evolution of supergenes underlying polymorphisms. This paper focuses on the social supergene in *Formica selysi*, where it is associated with queen numbers of colonies. Population data from mating flights, and colony sex ratio data is used to investigate the association between the supergene genotype and dispersal and mating behaviours. The data shows that the main difference between the social forms is not in the dispersal and swarming propensity, but in colony sex ratios, and assortative mating. The paper is important, clearly argued and analysed, and the results very interesting. I have some further analyses I would like to see though, as well as some clarifications.

1. As the sex allocation patterns are very interesting, it would be nice to see the colony and population data also presented and analysed as investment ratios. Presumably sexual dimorphism is not very high in this species, so the pattern should remain quite similar. Nevertheless, it would be an interesting complement to the big picture to understand the sex allocation patterns in this respect as well – whose optima is the sex ratio at, at the population level, or within colonies of either morph (and do the split sex ratio patterns make sense in this respect)?

Some further clarifications and explanations I would like to see:

2. A few sentences of explanation about the populations would be helpful: how closed are the populations? Are they clearly lineated patches of suitable habitat in a matrix of unsuitable habitat, or more continuous and less clearly defined? If the former, how far are the next populations? Some explanation on this would give the reader an idea of how exclusively the mating flight data corresponds to the colony data.

3. The possible scenario of hitch-hiking of P-males on MM-females, and the turning of M-nests into P via recruiting MP queens is very interesting. But given that such crossings are not found in the years of collections by the authors, some additional explanation would be helpful. This would mean that the colonies recruit new queens, and get rid of the old ones, before they are large enough to be found in field collections, right? But this would mean that they produce sexuals at very small colony sizes, which would be unusual, or that they recruit (or are infiltrated by) queens from the outside, which would be surprising, and would at least temporarily lead to multiple mtDNA lineages within colonies. The other option would be that for some reason the success of such crosses observed in lab rearings is not the case in the wild – if there are any likely scenarios for this, it would be useful to hear. So a few sentences of explanation would be nice.

Author's Response to Decision Letter for (RSPB-2021-0118.R0)

See Appendix A.

RSPB-2021-0118.R1 (Revision)

Review form: Reviewer 1

Recommendation

Accept as is

Scientific importance: Is the manuscript an original and important contribution to its field?

Excellent

General interest: Is the paper of sufficient general interest?

Excellent

Quality of the paper: Is the overall quality of the paper suitable?

Excellent

Is the length of the paper justified?

Yes

Should the paper be seen by a specialist statistical reviewer?

No

Do you have any concerns about statistical analyses in this paper? If so, please specify them explicitly in your report.

No

It is a condition of publication that authors make their supporting data, code and materials available - either as supplementary material or hosted in an external repository. Please rate, if applicable, the supporting data on the following criteria.

Is it accessible?

Yes

Is it clear?

Yes

Is it adequate?

Yes

Do you have any ethical concerns with this paper?

No

Comments to the Author

The authors have done an excellent job in answering the reviewer comments, and I have no further comments - I'm looking forward to seeing this published!

Decision letter (RSPB-2021-0118.R1)

31-Mar-2021

Dear Dr Fontcuberta

I am pleased to inform you that your manuscript entitled "Disentangling the mechanisms linking dispersal and sociality in supergene-mediated ant social forms" has been accepted for publication in Proceedings B.

Data Accessibility section

Open Access

Your article has been estimated as being 9 pages long. Our Production Office will be able to confirm the exact length at proof stage.

Paper charges

Sincerely,

Professor Gary Carvalho

Associate Editor:

Board Member: 1

Comments to Author:

Thank you for engaging so positively with the reviewer comments. The revised manuscript is improved as a result.

Board Member: 2

Comments to Author:

(There are no comments.)

Appendix A

Dear Editors,

We hope that the associate editor, editor-in-chief, and reviewers are in good health.

We thank you and the two reviewers for your efficient assessment of our MS, and we are very happy to have received positive feedback on it. We have now addressed all comments raised by the editor and referees. Detailed answers to specific questions are included below, with line numbers matching the new version.

We also include the revised manuscript with tracked changes at the end of this document.

We look forward to hearing from you in the near future.

All the best,

The authors

Associate Editor

Board Member: 1

Comments to Author:

Your manuscript has received two very positive reviews, with which I concur. As you'll see, there are some requests for clarification and additional text, which you need to address by making appropriate modifications in the manuscript. It's also important that you make the R code available, as highlighted by one of the reviewers. I look forward to the revised version and a cover letter explaining the additions and changes you've made.

We have done the appropriate modifications to the manuscript (see below). We added new analyses on sex investment and provided the R code.

Comments Referee: 1

Comments to the Author(s)

A social supergene system controls whether colonies of *Formica* ants have one or multiple queens. The authors perform extensive field collections and laboratory rearing experiments in order to understand the transmission of the supergene variants. This includes crosses and rearing of queens of the different genotypes with males of the genotypes, population-wide captures of alates leaving their nests, capturing and genotyping of male and queen brood from within nests, and post mating-flight capturing of mated queens and their sperm. The results read a bit dry - this is not a weakness but rather a necessary strength for fully reporting the different experimental results.

Thank you for these nice comments.

The methods are appropriate, with high sample sizes except when the biology was limiting (e.g., a lack of flying queens from multiple-queen-colonies). Intriguingly, the authors detect that multiple-queen colonies produce almost exclusively males. This differs substantially from single-queen colonies, where colonies tend to be specialised in producing either queens or males. The reasons for this are unknown. What could be the ecological reasons for this? Alternatively, could there be genetic factors affecting sex allocation? There is some evidence of supergene-related sex distortion in the fire ant system (a Ross paper, and Buechel et al Mol Ecol 2014).

We have added a paragraph on potential mechanisms explaining differences in sex allocation, and added the suggested references (lines 332-340).

Crucially (and somewhat counterintuitively), the results suggest that dispersal of polygyne colonies likely occurs through initial establishment of new colonies by monogyne queens, with subsequent conversion of these colonies to polygyny.

The results are of the quality and general relevance that are appropriate for Proc B. The paper stands well as is, thus I feel that I do not need to see a revised version of this paper.

Thank you for these nice comments.

Do the authors have any mitochondrial sequence data that could support the general conclusions of monogyne-queen-mediated polygyne founding?

We indeed have full genome sequence data indicating an absence of mitochondrial DNA differentiation between the monogyne and polygyne social form. These unpublished data are in accordance with female-mediated gene flow between social forms, but do not provide unequivocal evidence that monogyne queens found polygyne colonies, and discussing them would be beyond the scope of the current article.

I have mostly minor suggestions:

- Line 30: citation 15 is unnecessary, as citation 17 cited 2 lines later is the actual paper reporting this. I am also not convinced that an unpublished PhD thesis (citation 16) should be cited.

We have removed the citations as suggested.

- Line 73: grammar: "whether" is inappropriate

Thank you for pointing this out. We have changed whether for whereas.

- supergene genotyping: the authors develop and report a novel genotyping assay, but provide no rationale or mechanism through which it was developed. This likely includes some SNP comparisons? Does it target the same polymorphism as the previous RFLP assay? Could the ongoing low-level recombination observed in another of the authors recent papers be problematic for this?

The SNPs used in the qPCR assay differ from those targeted in the RFLP assay. We provide the TaqMan probes and added the rationale through which we developed them (lines 153-155).

The recombination/gene conversion events detected in Brelsford et al 2020 are extremely rare. They were detected when comparing distant species, but do not appear to cause within-species variation in *Formica selysi*.

- line 229 - given the paucity of polygyne queens (n = 6 and n=4, compared to 86 monogyne), I am not convinced that this conclusion wants to be drawn.

We agree that with such small sample size it is better not to build strong conclusions about whether polygyne queens are inseminated with the same frequency as queens from monogyne colonies. Moreover, this was not important for our conclusions. We have therefore removed this statistical comparison.

- The proc b reviewer form indicates "It is a condition of publication that authors make their supporting data, code and materials available" - the data appear to be here, but the code is not provided.

We have provided code and supporting data.

Referee: 2

Comments to the Author(s)

"Social chromosomes" of ants are an important model system in understanding the evolution of supergenes underlying polymorphisms. This paper focuses on the social supergene in *Formica selysi*, where it is associated with queen numbers of colonies. Population data from mating flights, and colony sex ratio data is used to investigate the association between the supergene genotype and dispersal and mating behaviours. The data shows that the main difference between the social forms is not in the dispersal and swarming propensity, but in colony sex ratios, and assortative mating. The paper is important, clearly argued and analysed, and the results very interesting. I have some further analyses I would like to see though, as well as some clarifications.

Thank you for these nice comments.

1. As the sex allocation patterns are very interesting, it would be nice to see the colony and population data also presented and analysed as investment ratios. Presumably sexual dimorphism is not very high in this species, so the pattern should remain quite similar. Nevertheless, it would be an interesting complement to the big picture to understand the sex allocation patterns in this respect as well – whose optima is the sex ratio at, at the population level, or within colonies of either morph (and do the split sex ratio patterns make sense in this respect)?

We have added extensive data and analyses on colony sex investment ratios, calculated by combining numerical sex-ratios with difference in dry weight between females and males (ESM). We find overall the same pattern (see lines 278-279 and ESM).

We have added a paragraph on sex allocation patterns, sex-ratio conflict and the underlying mechanisms, with reference to a previous study suggesting queen control of sex allocation in the monogyne social form (lines 331-340).

Some further clarifications and explanations I would like to see:

2. A few sentences of explanation about the populations would be helpful: how closed are the populations? Are they clearly linedated patches of suitable habitat in a matrix of unsuitable habitat, or more continuous and less clearly defined? If the former, how far are the next populations? Some explanation on this would give the reader an idea of how exclusively the mating flight data corresponds to the colony data.

We have added more information on the two populations sampled, the origin of females and males in swarms, and how they relate to the colony sex allocation data (lines 110-114).

3. The possible scenario of hitch-hiking of P-males on MM-females, and the turning of M-nests into P via recruiting MP queens is very interesting. But given that such crossings are not found in the years of collections by the authors, some additional explanation would be helpful. This would mean that the colonies recruit new queens, and get rid of the old ones, before they are large enough to be found in field collections, right?

That is correct. We usually sample large, old, mature colonies in the field, and we do not sample all queens, so detecting the founding MM queen mated to a P male is challenging.

But this would mean that they produce sexuals at very small colony sizes, which would be unusual, or that they recruit (or are infiltrated by) queens from the outside, which would be surprising, and would at least temporarily lead to multiple mtDNA lineages within colonies. The other option would be that for some reason the success of such crosses observed in lab rearings is not the case in the wild – if there are any likely scenarios for this, it would be useful to hear. So a few sentences of explanation would be nice.

We have extended the discussion on these possible explanations, which is now more balanced. We mention data showing that part of the incipient colonies headed by MM queens mated to P males may fail early on, prior to reaching maturity. In the laboratory, after a year these colonies were smaller and less competitive than colonies headed by MM queens mated to M males. But we also have data indicating that small incipient colonies with P-carrying workers accept P-carrying queens when they are accompanied by workers. We therefore think that it is likely that the few successful incipient colonies headed by MM queens mated to P males will recruit sister queens or queens originating from nearby polygyne colonies at a relatively early stage in colony development. We have expanded on this in the MS (lines 368-372 and lines 376-378).

In addition to the requested modifications, we have also improved the bins used in sex-allocation histograms (Figure 2 and Figure S3 (previous S2)) to make the x-axis clearer. We also modified Figure 2 to include colonies of the two populations, to better match with the new Figure S2.